# Declining Trend in Adolescent Alcohol Use: Does It Have Any Significance for Drinking Behaviour in Young Adulthood?

**DOI:** 10.3390/ijerph19137887

**Published:** 2022-06-27

**Authors:** Ingeborg Rossow, Inger Synnøve Moan, Elin K. Bye

**Affiliations:** Department of Alcohol, Tobacco and Drugs, Norwegian Institute of Public Health, 0213 Oslo, Norway; ingersynnove.moan@fhi.no (I.S.M.); elinkristin.bye@fhi.no (E.K.B.)

**Keywords:** alcohol use, adolescents, young adults, cohort, time trends

## Abstract

Since 2000, adolescent alcohol use has declined substantially in many high-income countries, particularly in Northern Europe. This study examined whether birth cohorts in Norway who experienced different levels of alcohol consumption in mid-adolescence differed in drinking behaviour when they reached young adulthood. We analysed data from annual population surveys in Norway (2012–2021). The analytic sample comprised data from respondents aged 20–29 years (N = 5266), and we applied four birth cohorts (i.e., 1983–1987, 1988–1992, 1993–1996 and 1997–2001). We applied age categories with two- and five-year intervals and tested whether drinking frequency, heavy episodic drinking (HED) and usual number of drinks per drinking occasion during the past 12 months differed by birth cohort in age-specific strata. Possible cohort differences within age groups were tested using Pearson’s Chi square. There were no statistically significant differences between cohorts with respect to drinking frequency or HED frequency. However, the youngest cohort had fewer drinks per occasion when in their early 20s compared to older cohorts. This study showed that birth cohorts who differed substantially in levels of alcohol consumption in mid-adolescence, only to a little extent differed in drinking behaviour in young adulthood.

## 1. Introduction

Since 2000, adolescent alcohol use has declined substantially in many high-income countries, particularly in Northern Europe [1]. The decline pertains to various measures of drinking behaviour: prevalence of past year drinking and past month drinking [1,2], and volume of consumption [3]. Several aspects of the declining trends have received research attention in recent years: (i) descriptions of trends across countries, demographics and drinking behaviours (e.g., [2]); (ii) explanations for why these trends occurred (e.g., [4]); and (iii) implications of the declining trends for drinking in young adulthood [5]. In this paper, we examine the latter issue, employing data from Norway. By way of study motivation, we will review the extant literature on this topic.

Adolescent alcohol consumption is correlated with consumption in young adulthood, and thus heavier consumption in adolescence is found to predict heavy consumption and alcohol problems in young adulthood [6,7,8]. Studies of drinking trajectories from adolescence into adulthood also suggest that the drinking level in adolescence most often is maintained or increased in young adulthood [9,10]. Moreover, the decline in the prevalence of drinking in early and mid-teenage years implies that the proportion having an early onset of drinking has also declined. As early onset of drinking is correlated with heavy drinking in young adulthood, it has been argued that delaying the onset of drinking may prevent heavy drinking in adulthood [7,11]. Although the evidence to support this claim seems weak [12,13], it is nevertheless suggested that the substantial reduction in drinking among adolescents may have carried over to young adults, implying potential public health gains [5]. To this end, few studies have empirically examined this issue. Analysing Australian data, Livingston and colleagues found that more recent birth cohorts (who experienced less drinking in adolescence) had lower levels of drinking in young adulthood; however, cohort differences vanished with increasing age and narrowed substantially from the age of 18 onward [14]. Lintonen and colleagues [15] examined whether the decrease in adolescent drinking continued into young adulthood in Finland but found no cohort differences in drinking at the age of 18.

In this study, we will add to the sparse literature on this topic by analysing data from Norway. The minimum legal age for purchasing alcohol is 18 years for beer, wine and other alcoholic beverages below 20% and 20 years for stronger alcoholic beverages. From the European School Project on Alcohol and Drugs (ESPAD), surveys among 15–16-year-old students in Norway have shown that the prevalence of past year drinking declined in a fairly linear fashion from 78% in 1999 to 45% in 2015, and the corresponding figures for the prevalence of past month drinking were 55% and 23%, respectively [16] (Figure 1). Correspondingly, the proportion reporting frequent drinking occasions were more than halved over the same period [16]. Finally, from 1999 to 2015, the proportion of adolescents reporting any heavy episodic drinking (5+ units of alcohol at one drinking occasion) declined from 50% to 16% [16]. By employing cross-sectional data for birth cohorts who experienced markedly different levels of alcohol consumption in mid-adolescence, i.e., roughly corresponding to the cohorts participating at age 15–16 years in the 1999 and 2015 ESPAD surveys, we will explore whether drinking behaviour differs by cohort in young adulthood.

## 2. Data and Methods

### 2.1. Samples and Data Collections

This study is part of a Nordic research collaboration project entitled *“Twenty years later: Explanations and consequences of the decline in adolescents’ drinking in the Nordic countries”* which was established in 2020. We employed data from the annual general population surveys on alcohol and drug use in Norway from 2012 through 2021 (average response rate 58.4%). The survey samples comprise adults aged 16–79 years from the general population in Norway. Sampling procedures, measurements and data collection methods are identical over time, and Statistics Norway conducts the surveys on behalf of the Norwegian Institute of Public Health. The surveys were conducted according to the guidelines of the Declaration of Helsinki, the Personal Data Act and the Statistics Act. Each survey year, respondents in the age group 16–25 years were oversampled. For the present study, we applied only data for respondents aged 20 through 29 years. The number of respondents by birth cohort and survey year is displayed in Table 1. The net data set comprised a total of 5266 respondents.

### 2.2. Measures

Age in years at survey date was recorded from the population registry, from which the survey samples were drawn. Birth cohort was calculated from age (continuous measure) and survey year, and we applied four birth cohorts (i.e., 1983–1987, 1988–1992, 1993–1996 and 1997–2001). Thus, the oldest cohort, born in the period 1983 to 1987, corresponds roughly to the cohort participating at age 15–16 years in the ESPAD surveys in 1999 and 2003 when alcohol use was most prevalent. The youngest cohort, born in the period 1997 to 2001, corresponds roughly to the cohort participating in the 2015 ESPAD survey, when alcohol use was far less prevalent among 15–16-year olds [16] (see Figure 1).

Respondents reported whether they had consumed any alcohol in the past 12 months, and if so, how frequently they had consumed alcohol. Among past year drinkers, responses to one of eight frequency options, from ‘once’ to ‘every day’ were used to construct a semicontinuous measure of past year drinking frequency. We categorized drinking frequency into three dichotomous measures of drinking behaviour: any drinking during the past 12 months, monthly drinking (i.e., at least once a month) and weekly drinking (i.e., at least once a week). Moreover, a frequency measure of heavy episodic drinking (HED) (i.e., drinking 5+ units of alcohol on one occasion) was also categorized into three dichotomous measures: (i) during the past 12 months, (ii) monthly and (iii) weekly. Finally, usual number of drinks per occasion (past 12 months) was measured, with response options 1–2, 3–4, 5–6, 7–9 and 10 or more. Notably, the risk of overall health harm increases with increasing volume of consumption [17] and with increasing HED frequency [18].

We included only respondents with valid responses on the measures of drinking frequency (missing observations = 2), HED frequency (missing observations = 89) and usual number of drinks per occasion (missing observations = 28). Hence, the analytic sample comprised a total of 5266 respondents.

### 2.3. Analytic Approach and Statistical Analyses

In the initial step, we visualized drinking behaviour by age and cohorts in diagrams. For these analyses, we applied age categories with two-year intervals. We excluded observations from cell counts < 100. Next, we tested whether drinking behaviour differed by birth cohort in age-specific strata. For these analyses, we grouped age into two categories: 20–24 years and 25–29 years to obtain larger cell counts and reduce random variation. We tested possible cohort differences within age groups, using Pearson’s Chi square. Statistical significance level was set to *p* < 0.05.

Although young people were over-sampled in the series of Norwegian surveys, we employed weighted data in the main analyses to achieve the most comparable samples possible.

## 3. Results

By visual inspection of diagrams, there was little difference between birth cohorts in drinking behaviour by two-year age groups in the span 20–29 years. The only exception was weekly drinking at age 20–21 years, for which we found a significantly lower proportion in the youngest cohort (Figure 2, Figure 3 and Figure 4).

Correspondingly, we examined the proportion reporting past year monthly and weekly heavy episodic drinking (HED) by age and birth cohort. As can be seen from Figure 5, Figure 6 and Figure 7, there was an overall tendency that HED prevalence decreased with increasing age, whereas no clear cohort differences were observed.

In a similar vein, we explored mean annual drinking frequency and usual number of drinks per occasion by age group and cohort (Figure 8 and Figure 9). Usual number of drinks per occasion tended to decrease with increasing age, and particularly in younger groups, significant cohort differences were observed, with lower intake per occasion in younger cohorts (Figure 9).

Next, we tested possible cohort differences in drinking behaviour within broader age groups (Table 2). There were no statistically significant differences between cohorts with respect to the prevalence of past year drinking, monthly drinking or weekly drinking. Correspondingly, no statistically significant cohort differences were observed within these broader age groups with respect to HED.

Moreover, a test of whether mean annual drinking frequency differed by cohort within the two age groups revealed no statistically significant difference (Table 3). With respect to usual number of drinks per occasion, however, a significant cohort difference was identified: the younger cohorts had on average fewer drinks per occasion than older cohorts.

## 4. Discussion

While a large number of studies have demonstrated a substantial decline in adolescent alcohol use in many high-income countries since 2000 [2,3,19,20,21,22,23,24,25,26], few studies have examined the implications of the declining trends among adolescents for alcohol use in young adulthood. This study extended this sparse literature by examining whether birth cohorts who experienced markedly different levels of alcohol consumption in mid-adolescence, differed by cohort in young adulthood. By employing a series of cross-sectional surveys among young adults in Norway, we found that the substantial cohort differences in drinking behaviour observed when these cohorts were in their mid-teens, to a little extent transferred into young adulthood. However, there was one exception from this overall picture. The youngest cohort who experienced far less alcohol use in their mid-teens, reported having fewer drinks per occasion when in their early 20s compared to older cohorts.

Our findings mirror those of the few previous studies on this topic [14,15], demonstrating that younger cohorts who experienced less drinking and HED when they were in their mid-teens did not differ much in drinking behaviour in young adult age. The differences pertained mainly to frequent and heavy drinking. These findings may thus suggest that the decline in alcohol use and heavy episodic drinking among 15–16-year-olds over the past two decades only to some extent impacted alcohol use among young adults.

In other words, when following drinking trajectories of cohorts (i.e., at the population level) over an age span, a very different pattern emerges compared to when we follow individual trajectories. At the population level, there seems to be a substantial convergence between cohorts in drinking behaviour with increasing age as judged from the extant literature. At the individual level, on the other hand, the trajectories—as described in previous studies—do not converge much, and the drinking level in mid-adolescence tends to predict drinking level in young adulthood [6,7,8,9,10]. This difference in trajectory pattern between population level and individual level can be explained in several ways. First, both cohort trajectories and individual trajectories are inherently confounded by period effects although in different ways. Cohort trajectories are the product of age, period and cohort, and differences between cohorts at given ages may well reflect period effects, for instance, in terms of changes in normative climate and in parenting practices and norms over time [2]. In Norway, parenting practices and parent–child relationships have changed over the past decades: parents monitor their adolescent children more closely [26], and adolescents spend more time at home with their parents [27]. Conversely, the interindividual differences observed in individual trajectories probably reflect, among other things, stable individual predictors of drinking behaviour, including genetics and personality characteristics such as sensation seeking [28]. Thus, for these reasons, individual level findings that modest drinking in adolescence seems to carry over into young adulthood do not necessarily transfer to the aggregate level.

Our findings indicate that younger cohorts ‘catch up’ with older cohorts when reaching young adulthood. This may suggest that the factors which have driven the substantial decline in adolescent drinking are mainly of importance during adolescence. This assumption is in line with some findings from previous studies. Parental rules, practices and norms seem to be among these factors, as they contribute to explain—in statistical terms—the decline in drinking among adolescents [2,20,27,29], and they are part of adolescents’ accounts of why they refrain from drinking [30]. Other factors that seem to explain some of the downward trend in drinking include a decline in perceived availability [26] and less time spent on hanging out with friends in the evening [27]. In Norway, there are no clear indications that any prevention programs or changes in alcohol policy may have contributed to the decline in adolescent drinking.

The public health implications of the decline in adolescent drinking have been discussed, both with respect to the immediate short-term implications as well as possible long-term effects [31]. There is some evidence that the decline in adolescent alcohol use was accompanied by reductions in alcohol-related harms in the short term [19,31,32], although this literature is still surprisingly meagre [14]. Whether there is a potential public health gain from lower consumption if continued into young adulthood remains to be seen. The findings of the present study suggest that these would, at best, be quite modest. However, if a similar pattern is also found in other countries, the impact on public health may—in sum—be large. It should also be noted that adolescents are particularly vulnerable to the effects of alcohol due to incomplete neurocognitive development [33]. Thus, a delayed onset of alcohol use among adolescents will undoubtedly have a positive impact on public health.

Several study limitations should be noted. First, we did not have comparable data series prior to 2012; hence we could not describe drinking behaviour of the older cohorts when they were in their early 20s. Correspondingly, it is not yet possible to track drinking behaviour of the youngest cohorts more than a few years into young adulthood. Thus, based on available data, we can only to a limited extent describe cohort trajectories over the age span from 20 to 30 years. The present analyses were also limited by statistical power to perform fine-grained analyses with respect to more narrow categories of birth cohorts and age groups. Finally, some general limitations with population surveys on alcohol consumption should be noted. Heavy drinkers are often under-represented in survey samples, and responses to questions about drinking behaviour are often inaccurate due to memory bias, problems in conceptualizing alcohol consumption and a tendency to downplay one’s own drinking (i.e., social desirability bias) [34]. Thus, our estimates of drinking prevalence and frequency are likely downward biased. There is, however, no obvious reason to assume that such bias would differ systematically with birth cohort or survey year to the extent that it impacted our results.

## 5. Conclusions

Research on the implications of the declining trends in alcohol use among adolescents for drinking in young adulthood is sparse. This study showed that birth cohorts in Norway who differed substantially in levels of alcohol consumption in mid-adolescence, only to a little extent differed in drinking behaviour as young adults. This suggests that the decline in adolescent drinking since 2000 mainly implied a delay of drinking onset. However, compared to older cohorts, the youngest cohort reported drinking less frequently and lower amounts of alcohol per occasion when in their early 20s. Thus, in addition to the potential health gain related to the delayed onset of drinking in adolescence, the youngest cohort may possibly also experience a positive health impact of keeping their consumption at a slightly lower level into adulthood than older birth cohorts.

## Figures and Tables

**Figure 1 ijerph-19-07887-f001:**
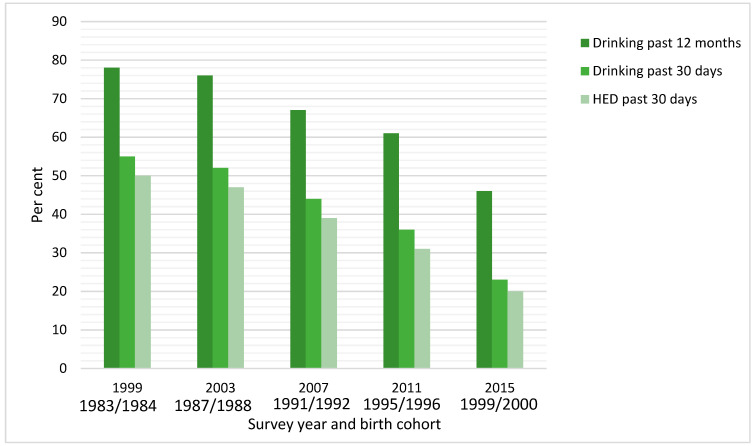
Prevalence of past year drinking, past 30 days drinking and past 30 days heavy episodic drinking (HED) by survey year and corresponding birth cohort. Data from the European School Project on Alcohol and Drugs (ESPAD) among 15–16 year olds in Norway. (Source: Norwegian Institute of Public Health).

**Figure 2 ijerph-19-07887-f002:**
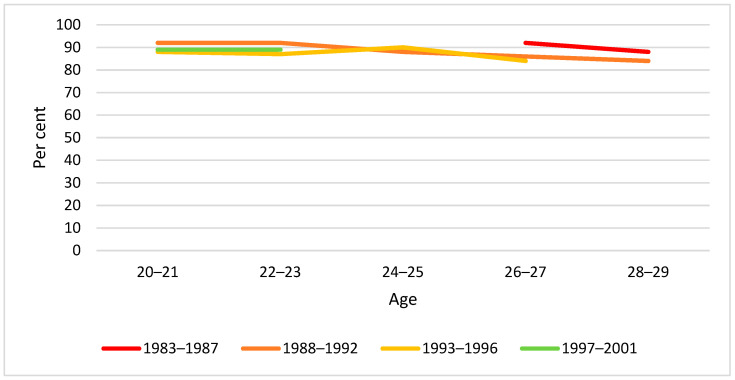
Proportion past year drinkers by birth cohort and age. Note: None of the cohort differences at any age were statistically significant (*p* > 0.05).

**Figure 3 ijerph-19-07887-f003:**
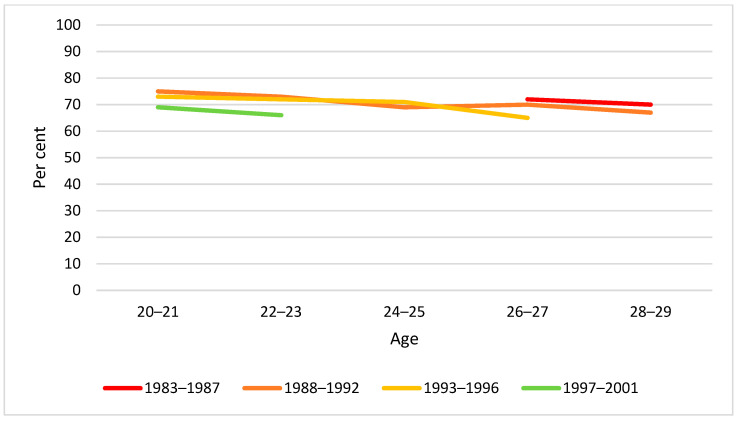
Proportion monthly drinking by birth cohort and age. Note: None of the cohort differences at any age were statistically significant (*p* > 0.05).

**Figure 4 ijerph-19-07887-f004:**
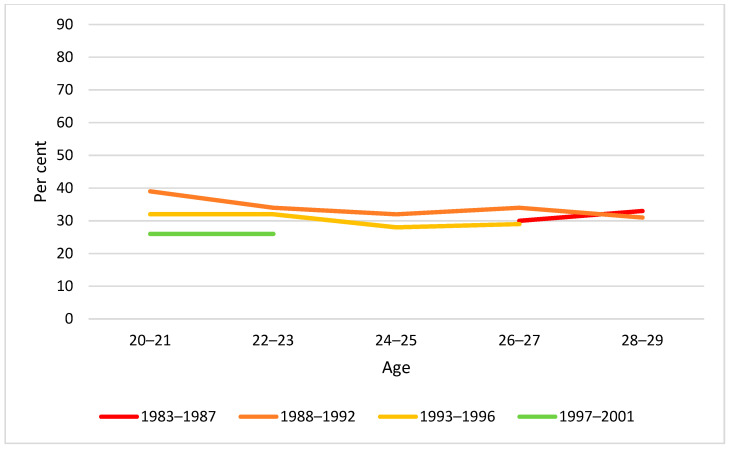
Proportion weekly drinking by birth cohort and age. Note: None of the cohort differences at any age were statistically significant (*p* > 0.05).

**Figure 5 ijerph-19-07887-f005:**
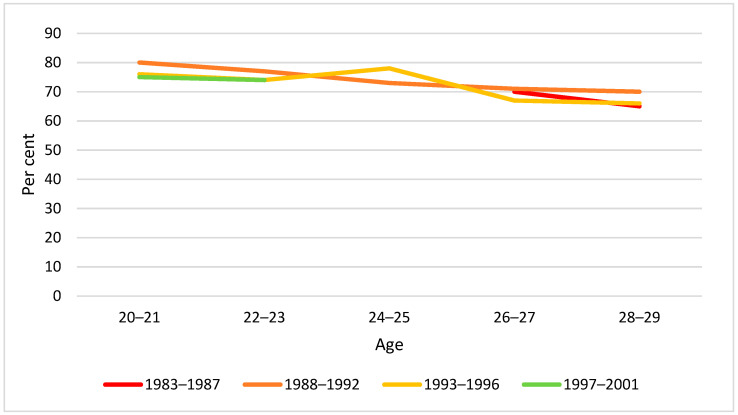
Past year HED prevalence by age and birth cohort. Note: None of the cohort differences at any age were statistically significant (*p* > 0.05).

**Figure 6 ijerph-19-07887-f006:**
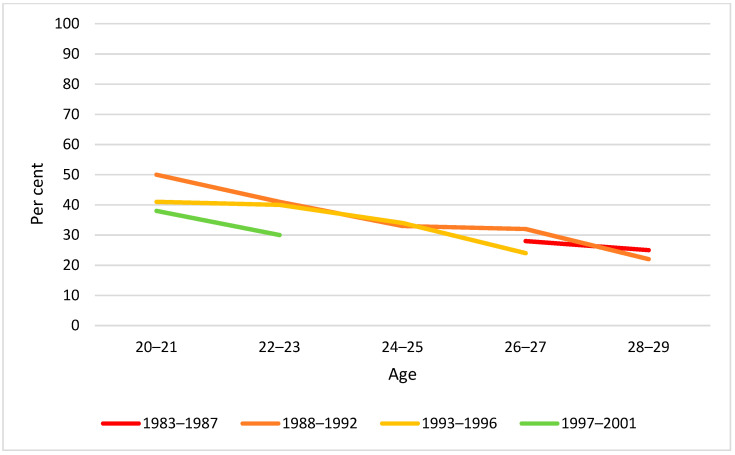
Proportion monthly HED by birth cohort and age. Note: None of the cohort differences at any age were statistically significant (*p* > 0.05).

**Figure 7 ijerph-19-07887-f007:**
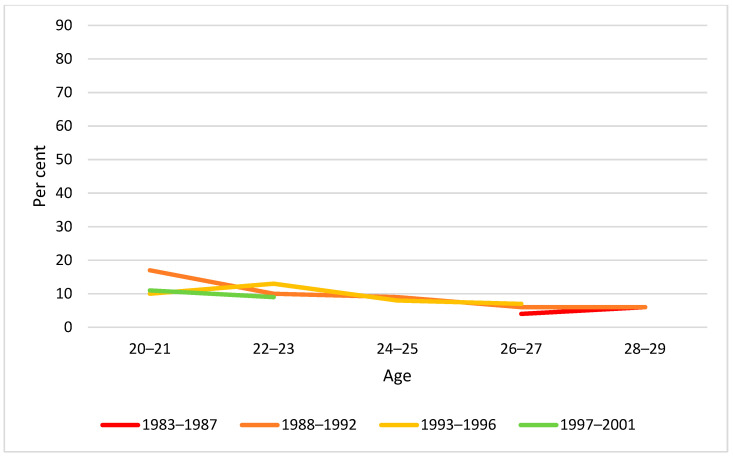
Proportion weekly HED by birth cohort and age. Note: None of the cohort differences at any age were statistically significant (*p* > 0.05).

**Figure 8 ijerph-19-07887-f008:**
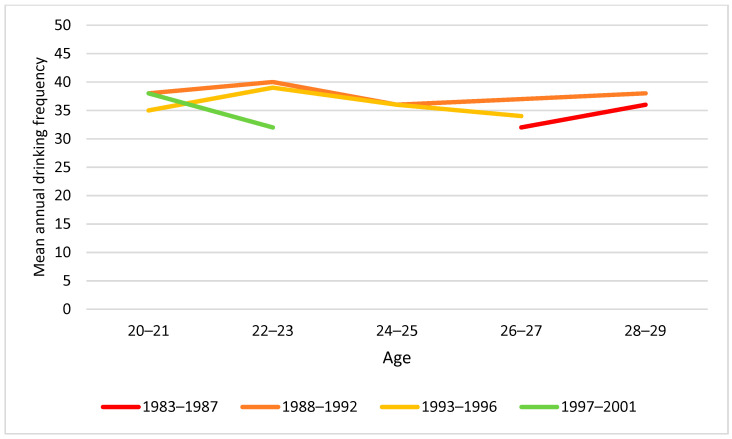
Mean annual drinking frequency by birth cohort and age. Note: None of the cohort differences at any age were statistically significant (*p* > 0.05).

**Figure 9 ijerph-19-07887-f009:**
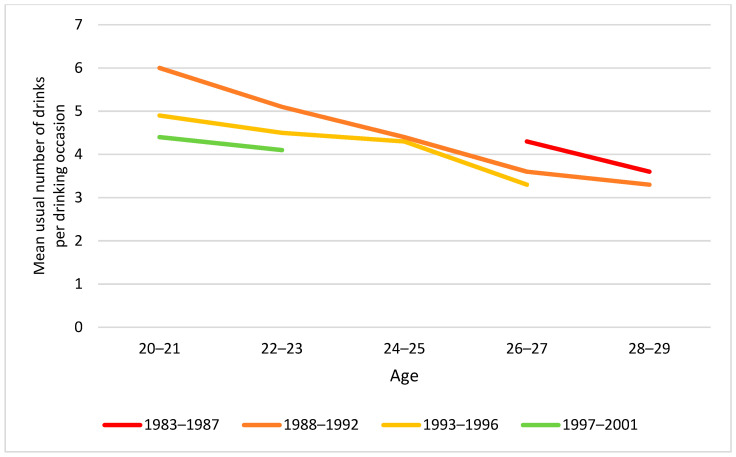
Mean usual number of drinks per drinking occasion by birth cohort and age. Note: At ages 20–21, 22–23 and 26–27, cohorts differed statistically significantly (respective *p*-values: *p* < 0.001; *p* = 0.009; *p* = 0.009). At ages 24–25 and 28–29, cohort differences were not statistically significant (*p* > 0.05).

**Table 1 ijerph-19-07887-t001:** Analytic sample size by birth cohort and survey year (unweighted data).

Cohort/Year	2012	2013	2014	2015	2016	2017	2018	2019	2020	2021	All Years
1983–1987	224	182	147	113	48	0	0	0	0	0	714
1988–1992	235	249	238	276	246	274	228	159	112	57	2074
1993–1996	0	55	102	172	200	204	222	217	216	236	1624
1997–2001	0	0	0	0	0	63	120	171	220	280	854
All cohorts	459	486	487	561	494	541	570	547	548	573	5266

**Table 2 ijerph-19-07887-t002:** Drinking behaviour prevalence by birth cohort and age group. Percent. Unweighted data.

Age Group/Birth Cohort	1985–1988	1989–1992	1993–1996	1997–2001	Chi Square Test
**Past year drinking**					
20–24 years	-	90.5	88.3	88.5	1.84, *p* = 0.398
25–29 years	88.8	85.3	84.8		4.36, *p* = 0.113
**Monthly drinking**					
20–24 years	-	67.8	65.3	62.5	3.43, *p* = 0.180
25–29 years	61.4	62.0	59.8		0.57, *p* = 0.754
**Weekly drinking**					
20–24 years	-	28.5	27.6	23.6	4.08, *p* = 0.130
25–29 years	28.2	28.4	25.8		0.94, *p* = 0.625
**Past year HED**					
20–24 years		75.7	75.0	73.9	0.51, *p* = 0.774
25–29 years	66.1	69.2	67.9		1.49, *p* = 0.476
**Monthly HED**					
20–24 years		39.6	38.5	33.8	4.61, *p* = 0.100
25–29 years	25.9	27.1	26.4		0.25, *p* = 0.882
**Weekly HED**					
20–24 years		11.3	10.8	9.4	1.19, *p* = 0.552
25–29 years	5.3	6.6	6.8		1.22, *p* = 0.544

**Table 3 ijerph-19-07887-t003:** Past year drinking frequency and usual number of drinks per occasion by birth cohort and age group. Means. Weighted data.

Age Group/Birth Cohort	1985–1988	1989–1992	1993–1996	1997–2001	F Test
**Drinking frequency**					
20–24 years	-	38.6	37.2	35.3	0.77, *p* = 0.461
25–29 years	35.0	36.8	35.3	-	0.35, *p* = 0.707
**Usual number drinks/occasion**					
20–24 years	-	5.1	4.6	4.3	9.31, *p* < 0.001
25–29 years	3.9	3.7	3.7		1.31, *p* = 0.270

## Data Availability

The data presented in this study are available on reasonable request to the corresponding author.

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
