# Peer review of "Declining Trend in Adolescent Alcohol Use: Does It Have Any Significance for Drinking Behaviour in Young Adulthood?"

_ijerph, 2022, doi:10.3390/ijerph19137887_

Round 1

Reviewer 1 Report

It is a remarkable paper about the declining patterns in adolescent alcohol use in Norway. However, several revisions are required as follows:

1. An ethical approval for the study should be additionally described.

2. Statistical significance of P-value should be additionally described.

3. The potential influence of severity level of alcohol use on the findings should be additionally discussed. 

Author Response

Thank you for the insightful and useful comments and suggestions regarding our manuscript. We have now revised the manuscript, taking into account the suggestions from the reviewers. A point-by-point response to Reviewer 1 follows below.

It is a remarkable paper about the declining patterns in adolescent alcohol use in Norway.

RE: We would like to thank Reviewer 1 for the positive comments regarding our manuscript.

However, several revisions are required as follows:

  1. An ethical approval for the study should be additionally described.

RE:  The surveys collected by Statistics Norway were conducted according to the guidelines of the Declaration of Helsinki, the Personal Data Act and the Statistics Act. This information has now been added on page 2, under chapter 2.1. Samples and data collections. We have also added the relevant information under the Institutional Review Board Statement and the Informed Consent Statement (see page 11).

  1. Statistical significance of P-value should be additionally described.

RE: We set the statistical significance level to p< .05. This is now added to chapter 2.3. Analytic approach and statistical analyses (see page 4).

  1. The potential influence of severity level of alcohol use on the findings should be additionally discussed. 

RE: We have now added a sentence to the Methods section (see page 3, paragraph 2) explaining the associations between alcohol consumption, HED and risk of health harms.

Reviewer 2 Report

In this article, the authors present data from the annual population survey in Norway (2012-2021). The sample comprised data from participants aged 20 through 29 years (N=5266).  The main objective of this article is to examine whether drinking behavior differ by cohort in young adulthood. The main findings showed that compared with older cohorts, the youngest cohort reported drinking less frequently and lower amounts of alcohol per occasion.  The manuscript is well written and covers an interesting and relevant topic for readers of IJERPH.

I suggest the authors to enrich the discussion. Between the possible reasons explaining the alcohol reduction in young people, they mentioned: “changes in normative climate and in parenting practices and other interindividual differences”. Please provide evidence that supports this asseveration.  I recommend extending the discussion, showing other structural factors that explain the reduction of alcohol consumption in young people in Norway. For example, if there were public policies oriented to the families informing about the risk of alcohol consumption in young people, or if the results are product of the international changes in substance use?.  

Minor observation.

It is important to specify if the research project was reviewed by an ethical committee, it is not specified how was the voluntary collaboration obtained. 

Author Response

Thank you for the insightful and useful comments and suggestions regarding our manuscript. We have now revised the manuscript, taking into account the suggestions from the reviewers. A point-by-point response to Reviewer 2 follows below.

Comments and Suggestions for Authors

In this article, the authors present data from the annual population survey in Norway (2012-2021). The sample comprised data from participants aged 20 through 29 years (N=5266).  The main objective of this article is to examine whether drinking behavior differ by cohort in young adulthood. The main findings showed that compared with older cohorts, the youngest cohort reported drinking less frequently and lower amounts of alcohol per occasion.  The manuscript is well written and covers an interesting and relevant topic for readers of IJERPH.

RE: We would like to thank Reviewer 2 for the positive comments regarding our manuscript.

I suggest the authors to enrich the discussion. Between the possible reasons explaining the alcohol reduction in young people, they mentioned: “changes in normative climate and in parenting practices and other interindividual differences”. Please provide evidence that supports this asseveration.  I recommend extending the discussion, showing other structural factors that explain the reduction of alcohol consumption in young people in Norway. For example, if there were public policies oriented to the families informing about the risk of alcohol consumption in young people, or if the results are product of the international changes in substance use?. 

RE:  We have now provided support for the claim that parenting practices and time spent with parents have changed over the past two decades in Norway (see page 10, paragraph 3). We have also added a bit on what we know about additional factors that are found to explain the declining trend, and furthermore that there are no indications that prevention programs or policy changes may have impacted on the trend (see page 10, paragraph 4).

Minor observation.

It is important to specify if the research project was reviewed by an ethical committee, it is not specified how was the voluntary collaboration obtained. 

RE: As previously described in a response to Reviewer 1, this information has been added under 2.1. Samples and data collections (see page 2) and under the Institutional Review Board Statement and the Informed Consent Statement (see page 3).

Reviewer 3 Report

The present manuscript aimed to explore whether birth cohorts in Norway who experienced different levels of alcohol consumption in mid-adolescence, differed in drinking behavior when they reached young adulthood.

The manuscript is clear but the analysis is elementary and the amount of analyzed data is very restricted. Following that, I have a problem with the novelty and significance of content for the broader readership.

The presented data can be of interest only to the limited circle of potential readers that are specifically interested in this particular subject.

Following the statistical analysis of the data, it seems that this study does not provide an advancement of the current knowledge.

The English language is appropriate and understandable.

The citation of references within the text of the manuscript as well as the citation of references in the reference list is not according to the Instructions for authors and should be corrected accordingly.

Author Response

Thank you for the insightful and useful comments and suggestions regarding our manuscript. We have now revised the manuscript, taking into account the suggestions from the reviewers. A point-by-point response to Reviewer 3 follows below.

The present manuscript aimed to explore whether birth cohorts in Norway who experienced different levels of alcohol consumption in mid-adolescence, differed in drinking behavior when they reached young adulthood.

The manuscript is clear but the analysis is elementary and the amount of analyzed data is very restricted. Following that, I have a problem with the novelty and significance of content for the broader readership. The presented data can be of interest only to the limited circle of potential readers that are specifically interested in this particular subject.

RE: We argue that the study findings are quite novel: only a couple of studies have to this end examined whether the decrease in adolescent drinking may have impacted on drinking in young adulthood. Thus, we state in the first para in the Discussion section (see page 9): “While a large number of studies have demonstrated a substantial decline in adolescent alcohol use in many high-income countries since 2000 (2, 3, 17-24), few studies have examined the implications of the declining trends among adolescents for alcohol use in young adulthood. This study extended this sparse literature [..]”. We believe the study is of interest to a broader readership, as the findings pertain to a substantial interest in the likely long-term consequences of the decline in adolescent drinking. Moreover, studies of this kind, examining longitudinal trajectories at the group level, are important for understanding time-dynamic changes in alcohol consumption. 

Following the statistical analysis of the data, it seems that this study does not provide an advancement of the current knowledge.

RE: It is not clear what the reviewer means by this comment. We argue, as above, that the study findings are quite novel given the few previous studies on this topic, and hence that they provide an advancement of current knowledge.

The English language is appropriate and understandable.

RE: We would like to thank Reviewer 3 for the positive comment regarding our manuscript.

The citation of references within the text of the manuscript as well as the citation of references in the reference list is not according to the Instructions for authors and should be corrected accordingly.

RE: The citation of references in the text (i.e., numbered references) is in accordance with Instructions to authors.